# Experimental Investigation of Thermal Passive-Reactive Sensors Using 4D-Printing and Shape-Memory Biopolymers

Francesco Cesarano [1,2], Marco Maurizi [1], Chao Gao [1], Filippo Berto [1], Francesco Penta [2] and Chiara Bertolin [1,*]

1 Department of Mechanical and Industrial Engineering, Norwegian University of Science and Technology, 7034 Trondheim, Norway
2 Department of Industrial Engineering, University of Naples Federico II, 80125 Naples, Italy
* Correspondence: chiara.bertolin@ntnu.no

**Abstract:** The introduction of 4D printing has revolutionized the concept of additive manufacturing; it is a promising technology that can bring immense advantages over classical production and manufacturing techniques, such as achieving programmed time-varying structures and consequently reducing production time and costs. The rise of 4D technology is considered an evolution of 3D printing due to the introduction of the fourth dimension: time. This is possible thanks to intelligent materials that can morph into programmed shapes in response to environmental stimuli, such as temperature, humidity, water, and light. When appropriately combined, these properties open the door to numerous applications in the engineering industry. This paper aims to provide information on the shape-memory effect (SME). To this scope, exploiting an already verified methodology available in the literature, a programmed flexural deformation were analyzed, experimentally changing the geometric parameters constituting the specimens. Experimental data were then processed to derive equations linking curvature to various independent parameters (such as temperature and time) through a quadratic and linear combination of the variables. This study contributes to a better understanding of current 4D-printing concepts through a mathematical characterization of the SME and its dependencies. In the study of the SME, such a complete methodological approach (analytical, experimental, and numerical) is a first step towards the design of more complex, bio-inspired components that could bring, in the coming years, development of passive sensors characterized by a combination of geometric properties that exploit a wider SME operating range to detect any variation of a physical quantity.

**Keywords:** 4D printing; shape-memory polymers; bioplastics; composite material; geometric parameters; sensors; sustainability

## 1. Introduction

Recent advances in 3D-printed sensors have opened new markets in various applications. These sensors can measure properties such as pressure, humidity, temperature, force, and displacement. Advanced 3D materials are enabling the industry to rapidly design and produce reliable, accurate, and cost-effective sensors to meet the needs of the food and pharmaceutical industry, environmental monitoring and biomedical applications, renewable energy, and robotics [1]. Primarily, sensors can be classified as active and passive sensors. Active sensors require an energy source to detect changes in the physical environment. Passive sensors receive energy from the outside or are stimulated by interactions with their surroundings [1]. Additive manufacturing was initially developed for rapid prototyping; nonetheless, the development and the investigation of these technologies and advanced materials (as smart materials) has enabled a rapid transition to 3D-printed sensing material systems [2,3]. Numerous active and passive sensors have been fabricated through additive manufacturing and are used in various industrial applications, e.g., in consumer electronics, automotive, IT, and telecommunications [4–7].

Active sensors: A wide range of active sensors have been realized; for example, in Saari et al., 2016 [8], fiber encapsulation additive manufacturing (FEAM) and thermoplastic elastomer additive manufacturing (TEAM) were combined to create a capacitive force sensor. The sensor consists of a 3D-printed rigid structure with wires embedded in a spiral pattern that emulates a flat-plate capacitor and thermoplastic helix, achieving a near-linear response of (emulated) capacitance change compared to the applied load. In Hong et al., 2019 [9,10], a pressure sensor was proposed, based on a fiber Bragg grating (FBG), manufactured using the FDM$^{TM}$ technique, i.e., through this technique, the structure of an FBG sensor was realized successfully by incorporating it into PLA without sacrificing sensing performance compared to commercial sensors. Another relevant example is described by Kwok et al., 2017 [11], who proposed the fabrication, characterization, stress testing, and application of a low-cost thermoplastic conductive composite that has been transformed into filament for 3D printing. They then prototyped a plastic thermometer to highlight the potential for sensing applications using this new filament. In Maurizi et al., 2019, FDM$^{TM}$ 3D-printed embedded strain sensors were proven to perform dynamic measurements under cyclic loading conditions [12]. These listed are only a few examples of active sensors that can be realized by additive manufacturing; however, as can be seen, they require components outside 3D production and sometimes even complex combination procedures. Some examples of passive sensors are therefore listed below.

Passive sensors: Unlike active sensors, passive sensors made by additive manufacturing do not use commercially available electronic cores and do not involve any power supply, thus making them highly innovative and more complex to design. For instance, given the complex requirements of gas monitoring under harsh conditions, Zhou et al., 2020 [13] presented a study on body-centered periodical mullite-based ammonia sensors fabricated by 3D printing for highly reliable ammonia detection; through ceramic lattice structures with a deposition of polyaniline mixed with Ag nanoparticles, the sensors achieved excellent mechanical performance, ensuring stable operation under stress. Kisic et al., 2020 [14] conducted an additive manufacturing study to realize a low-cost force sensor. The proposed sensor consists of an inductor, an elastic spacer, and a smooth ferrite plate. Both the inductor and spacer were additively manufactured, while commercially available ferrite material was used as the magnetic part of the sensor. The operating principle of the sensor is based on inductance variations, and the proposed sensor was designed, fabricated, and finally characterized over a force range of 0 to 2 N. A final exciting result that is reported is proposed in Dharmarwardana et al., 2018 [15]; they carried out in-depth analyses of organic crystals such as naphthalene crystals associated with homopolymer organic filaments that can be combined with 3D printing to make temperature sensors by exploiting the thermochromic properties of the crystals.

A 4D-printing technology overview: The proposed examples show the difficulty and complexity of combining appropriate materials with printing techniques to create reactive components such as active and passive sensors. This article proposes using 4D printing to make simple passive sensors, as 4D printing adds the time dimension to the 3D-printing process by exploiting the shape-memory effect (SME), which allows the programmed deformation of manufactured components. This will provide vitality to the design of shape-memory materials (e.g., PLA), using an external stimulus (such as heat and humidity) to trigger the object's transformation into another programmed structure. In detail, 4D printing is a combination of four primary variables: the type of technology used for printing (e.g., fused deposition modeling or stereolithography), the chosen material, the stimulus applied, and the programming parameters (such as the one-way or two-way SME). The right combination of these four parameters makes it possible to obtain 4D components with entirely different properties which can be used in various sectors and for different types of applications [15–24]. Different papers inspired this research; for example, in the work of Yu et al., 2020 [25] and van Manen et al., 2017 [15], thermal stress-induced deformations were analyzed for PLA components and carbon fibers composite PLA parts obtained from 3D printers to evaluate the variation of the SME. Based on this analysis, the authors then

proposed self-assembling structures for various applications. Bodaghi et al., 2019 [26] proposed the design and analysis of complex structures with self-morphing characteristics using 4D-printing technology, exploiting an FEA tool to replicate their thermo-mechanical behaviors. This literature highlights that 4D printing is an accessible technology with high potential in manufacturing various engineering components, such as safety devices, programmable structures, actuators, and sensors.

*Research Objectives and Structure of the Paper*

The present work is a natural continuation of the work presented in Cesarano et al., 2022 [27], which aimed to find a suitable experimental method to study the SME associated with viscoelastic behavior by analyzing the response of the smart material to a homogeneous thermal increase stimulus through various time–temperature combinations and different programming parameters. The smart material was the PLA, and the additive manufacturing technology was the FDM$^{TM}$ . Consequently, this study aims to exploit the one-way SME phenomenon to achieve controllable and programmable bending deformation modifying temperature (selected stimulus), time, and geometric parameters. In detail, after the introduction (Section 1) and a brief overview of the materials and method adopted (Section 2), the results and discussion (Section 3) show the flexural curvatures obtained from programmed 4D-printing processes as the programming parameters change. These numerical results are then combined to analyze the curvature dependency, which is the result of the SME, in relation to all the considered parameters. This preliminary study has also been conducted to lay a mathematical/analytical foundation to describe the shape-memory effect (of the PLA in this case) on which this technology is based. Indeed, experimental demonstrations and concepts of various kinds that give insight into the hidden potential of 4D printing are widely present in the literature; however, there are no data collections through a large number of experiments, which can lead precisely to inherent mathematical considerations, as in this research. A final important objective of the study is to highlight the possible applicability of SMPs that will be materials for engineered components that exploit SME specifically. As shown in previous research [17,25], this technology could be applied to the realization of self-assembling structures and thus conceptually to actuators. However, a more immediate field of application, not at all currently associated with 4D printing, is the sensor industry, which would match well with the properties of biopolymers such as PLA [28,29].

## 2. Materials and Methods

In this work, test specimens made of PLA were fabricated using an FDM$^{TM}$ Prusa MK3S 3D printer. Using a gravitational convection oven (Fisherbrand), the test specimens were then subjected to a homogeneous thermal stimulus to trigger the SME. More in detail, bi-layer plates with a rectilinear fill pattern were used as test specimens, with fibers arranged at 0° for one layer (along the most extended specimen dimension) and 90° for the other layer to achieve flexural deformation (reference specimen) [17,25]. The test specimens, with a reference size of 50 × 10 × 0.4 mm and a fill density of 100%, were placed into the oven only after it reached the target set temperatures (i.e., 60 °C, 70 °C, and 80 °C, respectively) and they were kept at that set temperature for a few minutes (i.e., 3 min, 5 min, and 10 min). This method was extensively validated and chosen to avoid high deformations caused by too-long exposure times. In fact, from preliminary tests, it was observed that high deformations were visible when specimens were exposed to a thermal increase from room temperature to the target temperature, due to their viscoelastic behavior. The selected method has allowed a wider range of results to be compared too [27]. For more details on the printing and testing parameters, the reader is referred to Cesarano et al., 2022 [27]. In total, 180 specimens suitable for the various experimental combinations were produced and analyzed to carry out this study. To validate the experimental results, each experiment was repeated three times, varying the selected geometric parameters under each time–

temperature combination. The following Equation (1) was used to calculate the flexural curvature ($\kappa$) and to measure and catalogue the obtained curvatures during the experiments:

$$\kappa = \left(2sin\left[tan^{-1}(\delta/x)\right]\right)/\sqrt{x^2 + \delta^2} \tag{1}$$

where $\delta$ is the maximum deflection obtained from the bending acting on the component, and $x$ is the abscissa to which the maximum deflection corresponds. Lastly, taking advantage of this, curve- and surface-fitting tools were used to analyze and combine all the results obtained (Figure 1).

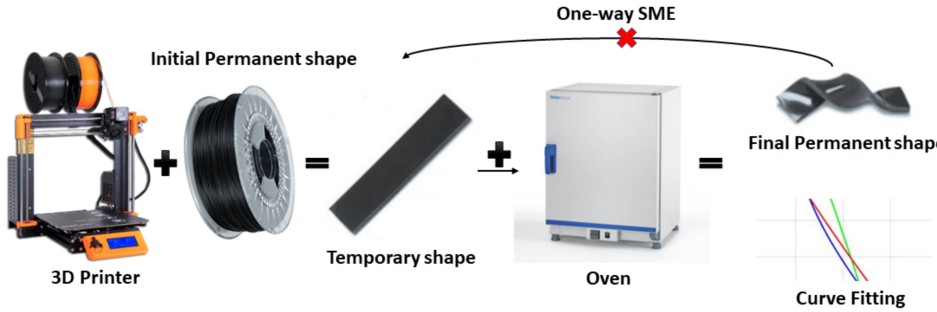

**Figure 1.** Outline of the experimental and analysis process (One-way SME + curve fitting).

## 3. Results and Discussion

The focus of the present research is on the analysis of the curvature variation with variation in the geometrical and printing parameters of the test specimens. The parameters concerned are the fill angle, the fill density, the thickness of each layer, and the total thickness of the specimen.

### 3.1. Fill-Angle Variation

The first geometrical parameter analyzed was the fill angle; therefore, the samples used had constant thickness (in total 0.4 mm) and density, and consisted of two layers with a density of 100% and a thickness of 0.2 mm per layer. Figure 2 shows the three types of layers used for this analysis; in detail, three specimens (a, b, and c) are shown with the following fill angles for the first and second layer: 0–45°, 0–60°, and 0–90°.

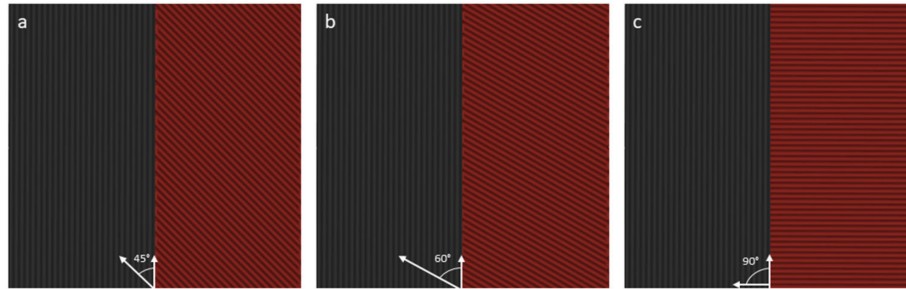

**Figure 2.** Fill-angle variation. (**a**) 0–45°; (**b**) 0–60°; (**c**) 0–90°.

Subsequently, the printed samples were subjected to the target temperatures (60 °C, 70 °C, and 80 °C) for a predetermined exposure time (3 and 5 min). Figure 3 shows the various samples subjected to different temperatures (first column = 60 °C; second column = 70 °C; third column = 80 °C) at exposure times of 3 (Figure 3a) and 5 min (Figure 3b). In both cases, up to a temperature of 70 °C, the deformation of the various specimens could be considered with good approximation to be exclusively flexural. At 80 °C, torsional deformation started to become more evident, as expected [17,25]. Probably, at 80 °C the complete release of the residual stresses took place and consequently the difference in ultimate deformation between the various specimens was more evident.

For example, the specimen with fibers arranged at 0° and 60° was the one that underwent greater torsional deformation, to the detriment of flexural deformation; in the specimen with fibers arranged at 0° and 90°, there was no torsional deformation, as already seen; finally, the one with fibers at 0° and 45° could be considered a combination of the two (Figure 3, top row). As studied in the literature [25], the torsional effect can be amplified by considering a constraint layer with fibers arranged at 90° and an actuation layer with fibers arranged at 45°. Table 1 shows the curvature values calculated with the geometric equation presented in the previous paragraph (Equation (1)).

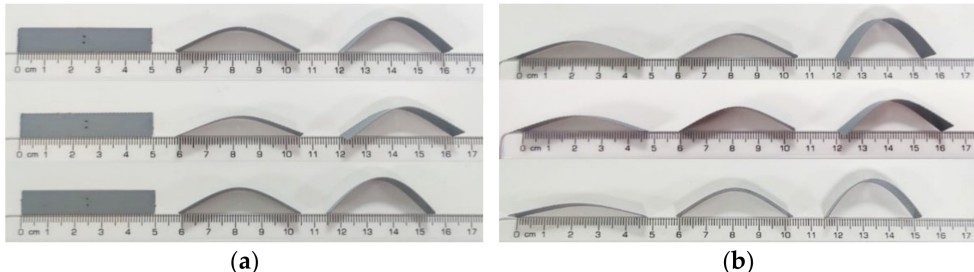

|  | (a) | | (b) |

**Figure 3.** Fill-angle variation (0–45° top, 0–60° center, 0–90° bottom row of the illustration). (**a**) 3 min; (**b**) 5 min.

**Table 1.** Curvature–fill-angle (FA) variation depending on the exposure time and temperature.

| Temperature Curvature (FA; Time) | T = 60 °C | T = 70 °C Values | T = 80 °C |
|---|---|---|---|
| $\kappa$ (0–45°; t = 3 min) | 0.0024 | 0.0301 | 0.0434 |
| $\kappa$ (0–60°; t = 3 min) | 0.0024 | 0.0256 | 0.0410 |
| $\kappa$ (0–90°; t = 3 min) | 0.0029 | 0.0249 | 0.0471 |
| $\kappa$ (0–45°; t = 5 min) | 0.0233 | 0.0319 | 0.0570 |
| $\kappa$ (0–60°; t = 5 min) | 0.0214 | 0.0364 | 0.0486 |
| $\kappa$ (0–90°; t = 5 min) | 0.0098 | 0.0311 | 0.0587 |

Hence, as can be seen from Figure 3 and Table 1, the curvature results show that twisted specimens exhibited a lower final bending curvature than the standard 0–90° specimen for high temperatures, while for temperatures close to the glass transition temperature, the 0–45° and 0–60° specimens exhibited higher curvature. This result may give a qualitative understanding that by exploiting different fill angles, a higher sensitivity of SME activation can be obtained. The variation of the fill angle was useful to show once again the presence of the shape-memory effect, depending on the initial programming of the specimen.

Due to the presence of torsional deformations, the validity of the geometric formula (1) derived from the bi-material strip theory [27] for calculating flexural curvature was compromised. Therefore, given the risk of noncomparability, it was decided not to investigate it further, limiting the number of experiments and not performing them for the 10 min exposure time (Figure 4).

### 3.2. Thickness Variation

The second parameter studied was the specimen thickness. Specimens with a filling density of 100%, with fibers arranged at 0° and 90°, with three different thicknesses, i.e., 0.2 mm, 0.4 mm, and 0.6 mm, were considered, with the two layers having the same thickness. As in the previous case, the same experimental method was used (each experiment was repeated three times); Figure 5 shows the various samples subjected to different temperatures (first column = 60 °C; second column = 70 °C; third column = 80 °C) at exposure times of 3 (Figure 5a), 5 (Figure 5b), and 10 min (Figure 5c).

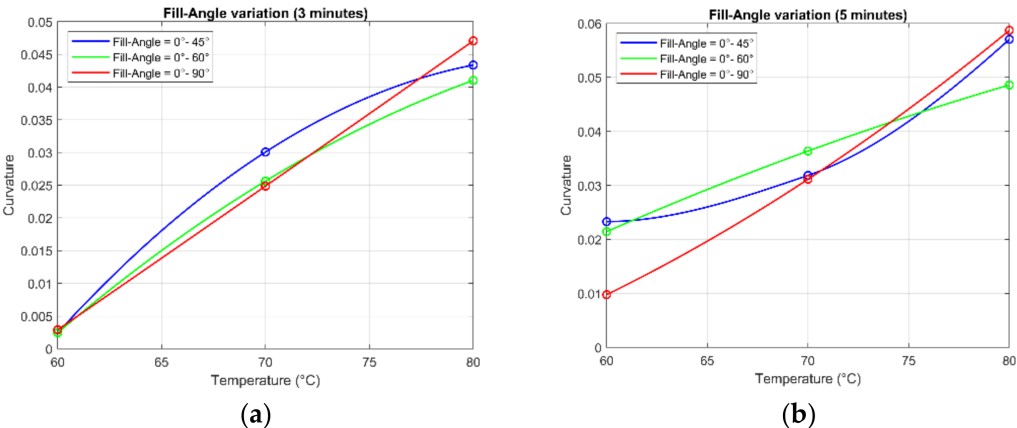

**Figure 4.** Curvature–temperature plot for fill-angle variation (0–45° blue; 0–60° green; 0–90° red) for (**a**) 3 and (**b**) 5 min.

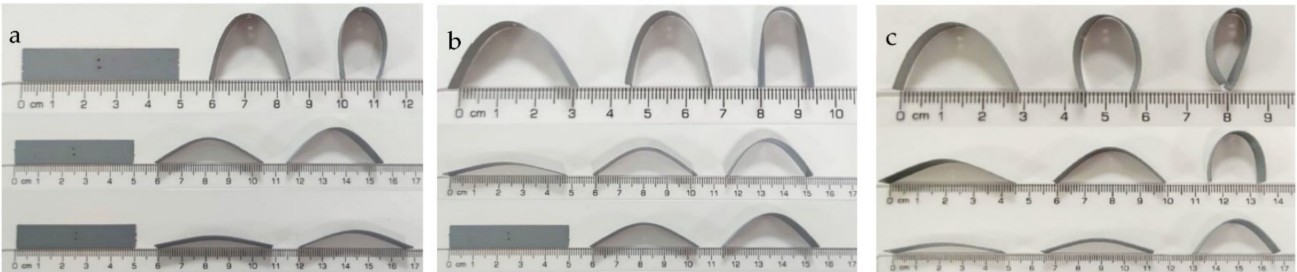

**Figure 5.** Experiment: thickness variation (0.2 mm—top, 0.4 mm—center, and 0.6 mm—bottom). (**a**) 3 min; (**b**) 5 min; (**c**) 10 min.

In Figure 5, there is a high increase in curvature as the thickness decreases; however, it can also be seen that the specimens with a total thickness of 0.2 mm (first row) for high time–temperature values (e.g., 10 min at 80 °C) show irregularities in the final shape. This finding suggested the possibility of a critical thickness condition, such that the combination of stresses due to SME and viscoelastic behavior caused local failure. As a result, the programmed deformation is not obtained but instead a component to be discarded. Table 2 reports the average curvature values obtained in all experiments, and the associated standard deviation (SD) values. The curvature–temperature diagrams for each exposure time analyzed are shown in Figure 6.

**Table 2.** Curvatures (mean values) and standard deviation—thickness (TK) variation depending on the exposure time and temperature.

| Temperature<br>Curvature (TK.; Time) $\pm$ SD | T = 60 °C | T = 70 °C<br>Values | T = 80 °C |
|---|---|---|---|
| $\kappa$ (0.2; t = 3 min) $\pm \sigma$ | $0.0093 \pm 0.0018$ | $0.0761 \pm 0.0004$ | $0.0913 \pm 0.0008$ |
| $\kappa$ (0.4; t = 3 min) $\pm \sigma$ | $0.0029 \pm 0.0004$ | $0.0248 \pm 0.0012$ | $0.0470 \pm 0.0013$ |
| $\kappa$ (0.6; t = 3 min) $\pm \sigma$ | $0.0006 \pm 0.0003$ | $0.0146 \pm 0.0019$ | $0.0251 \pm 0.0016$ |
| $\kappa$ (0.2; t = 5 min) $\pm \sigma$ | $0.0264 \pm 0.0006$ | $0.0803 \pm 0.0007$ | $0.0976 \pm 0.0024$ |
| $\kappa$ (0.4; t = 5 min) $\pm \sigma$ | $0.0097 \pm 0.0013$ | $0.0311 \pm 0.0032$ | $0.0587 \pm 0.0034$ |
| $\kappa$ (0.6; t = 5 min) $\pm \sigma$ | $0.0046 \pm 0.0013$ | $0.0178 \pm 0.0005$ | $0.0344 \pm 0.0027$ |
| $\kappa$ (0.2; t = 10 min) $\pm \sigma$ | $0.0672 \pm 0.0011$ | $0.0882 \pm 0.0050$ | $0.0919 \pm 0.0011$ |
| $\kappa$ (0.4; t = 10 min) $\pm \sigma$ | $0.0165 \pm 0.0032$ | $0.0413 \pm 0.0056$ | $0.0813 \pm 0.0011$ |
| $\kappa$ (0.6; t = 10 min) $\pm \sigma$ | $0.0120 \pm 0.0006$ | $0.0223 \pm 0.0005$ | $0.0561 \pm 0.0011$ |

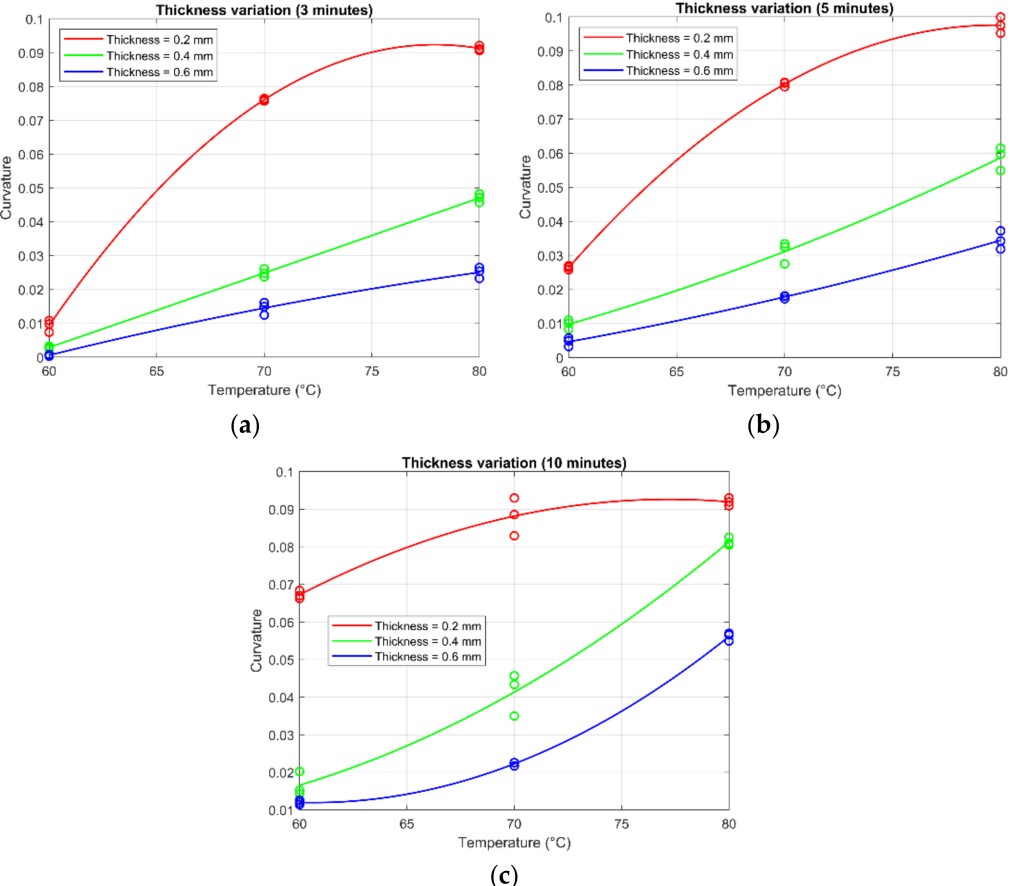

**Figure 6.** Curvature–temperature plot for thickness variation (0.6 mm blue; 0.4 mm green; 0.2 mm red) for (**a**) 3, (**b**) 5, and (**c**) 10 min.

The plots displayed (Figure 6) were obtained by curve fitting using a second-degree polynomial approximation. This approximation was chosen for several reasons.

First, this best fit works for all curvature–temperature points for any exposure time; moreover, given its simplicity, it returns a 95% goodness-of-fit rate. The equation that then approximates any combination of curvature, temperature, and time is as follows:

$$\kappa(T) = p_1 T^2 + p_2 T + p_3 \tag{2}$$

where $p_1$, $p_2$, and $p_3$ are fitting constants, $T$ is the temperature in degrees Celsius, and $\kappa$ is the curvature. Due to the simplicity of the equation, however, for each exposure time and thickness value, there are different fitting constants; consequently, in the following sections, one of the other examined two variables (thickness and time) is introduced into the fitting process, and thus into the equation, to continuously assess their variability, resulting in 3D plots. The following section describes a similar analysis corresponding to the fill-density variation.

### 3.3. Fill-Density Variation

The third and last parameter analyzed was the filling density. Two-layer specimens with fibers arranged at 0° and 90° were used, with a thickness of 0.2 mm for each layer (reference specimen). The variation of this parameter was studied by subjecting specimens with different fill densities to the same experiments, keeping them with identical dimensions between the two layers that make up the specimen. The fill densities were 60%, 80%, and 100%, as shown in Figure 7. A generic sample with 60% fill density is shown in Figure 8. The analysis is practically identical to the previous one; however, as can be seen from Figure 9, none of the specimens analyzed deformed irregularly, as in the case of the

0.2 mm-thick specimen; at the same time, a very pronounced curvature was obtained for a fill density of 60%.

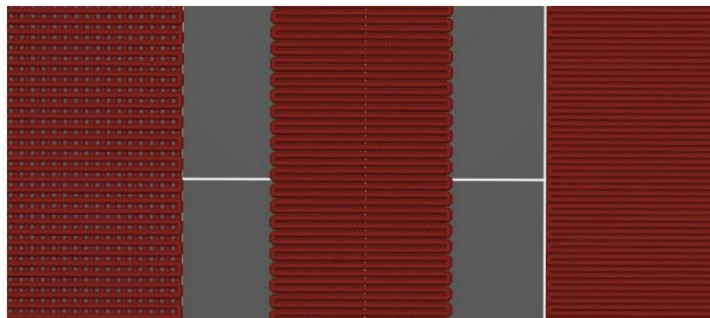

**Figure 7.** Fill density = 60% (**left**), 80% (**center**), 100% (**right**).

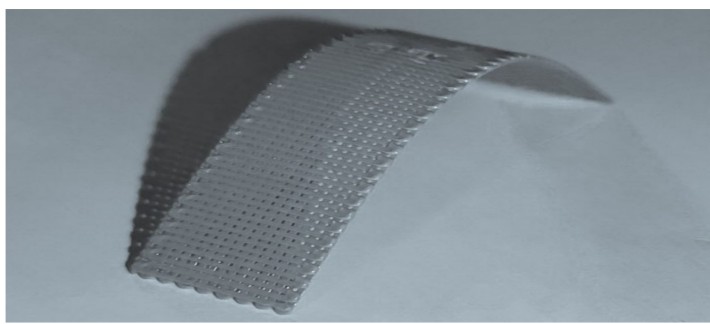

**Figure 8.** Generic sample (fill-density variation).

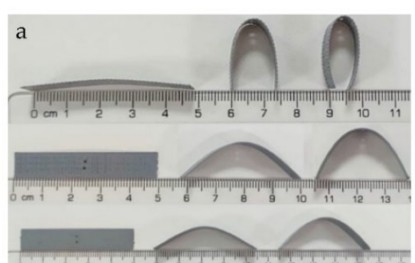 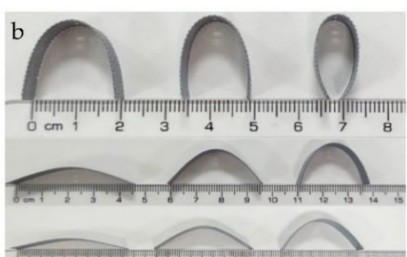 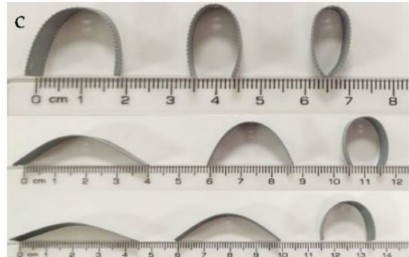

**Figure 9.** Experiment: FillDensity variation (60%—top, 80%—center, 100%—bottom). (**a**) 3 min; (**b**) 5 min; (**c**) 10 min.

This is a promising result for optimizing the activation range in relation to time and temperature because for temperatures close to the glass transition temperature (55 °C) and for low exposure times, greater curvatures are achieved than the thickness variation. Hence, Figure 9 shows the various samples subjected to different temperatures (first column = 60 °C; second column = 70 °C; third column = 80 °C) at exposure times of 3 (Figure 9a), 5 (Figure 9b), and 10 min (Figure 9c). As with the previous analysis, Table 3 also reports the results, containing the curvature obtained by the arithmetic mean of the three experiments' repetitions, and the associated standard deviation (SD) values. The curvature–temperature plots for each analyzed exposure time are shown below and reported in Figure 10.

**Table 3.** Curvatures (mean values) and standard deviation—fill-density (FD) variation depending on the exposure time and temperature.

| Temperature Curvature (FD.; Time) $\pm$ SD | T = 60 °C | T = 70 °C Values | T = 80 °C |
|---|---|---|---|
| $\kappa$ (60%; t = 3 min) $\pm \sigma$ | $0.0166 \pm 0.0017$ | $0.0831 \pm 0.0008$ | $0.0897 \pm 0.0020$ |
| $\kappa$ (80%; t = 3 min) $\pm \sigma$ | $0.0057 \pm 0.0009$ | $0.0480 \pm 0.0009$ | $0.0661 \pm 0.0028$ |
| $\kappa$ (100%; t = 3 min) $\pm \sigma$ | $0.0029 \pm 0.0004$ | $0.0281 \pm 0.0034$ | $0.0471 \pm 0.0013$ |
| $\kappa$ (60%; t = 5 min) $\pm \sigma$ | $0.0481 \pm 0.0019$ | $0.0907 \pm 0.0003$ | $0.0916 \pm 0.0007$ |
| $\kappa$ (80%; t = 5 min) $\pm \sigma$ | $0.0125 \pm 0.0012$ | $0.0545 \pm 0.0015$ | $0.0721 \pm 0.0007$ |
| $\kappa$ (100%; t = 5 min) $\pm \sigma$ | $0.0097 \pm 0.0013$ | $0.0311 \pm 0.0032$ | $0.0587 \pm 0.0034$ |
| $\kappa$ (60%; t = 10 min) $\pm \sigma$ | $0.0802 \pm 0.0011$ | $0.0892 \pm 0.0011$ | $0.0914 \pm 0.0006$ |
| $\kappa$ (80%; t = 10 min) $\pm \sigma$ | $0.0440 \pm 0.0016$ | $0.0673 \pm 0.0006$ | $0.0891 \pm 0.0005$ |
| $\kappa$ (100%; t = 10 min) $\pm \sigma$ | $0.0165 \pm 0.0032$ | $0.0413 \pm 0.0056$ | $0.0813 \pm 0.0011$ |

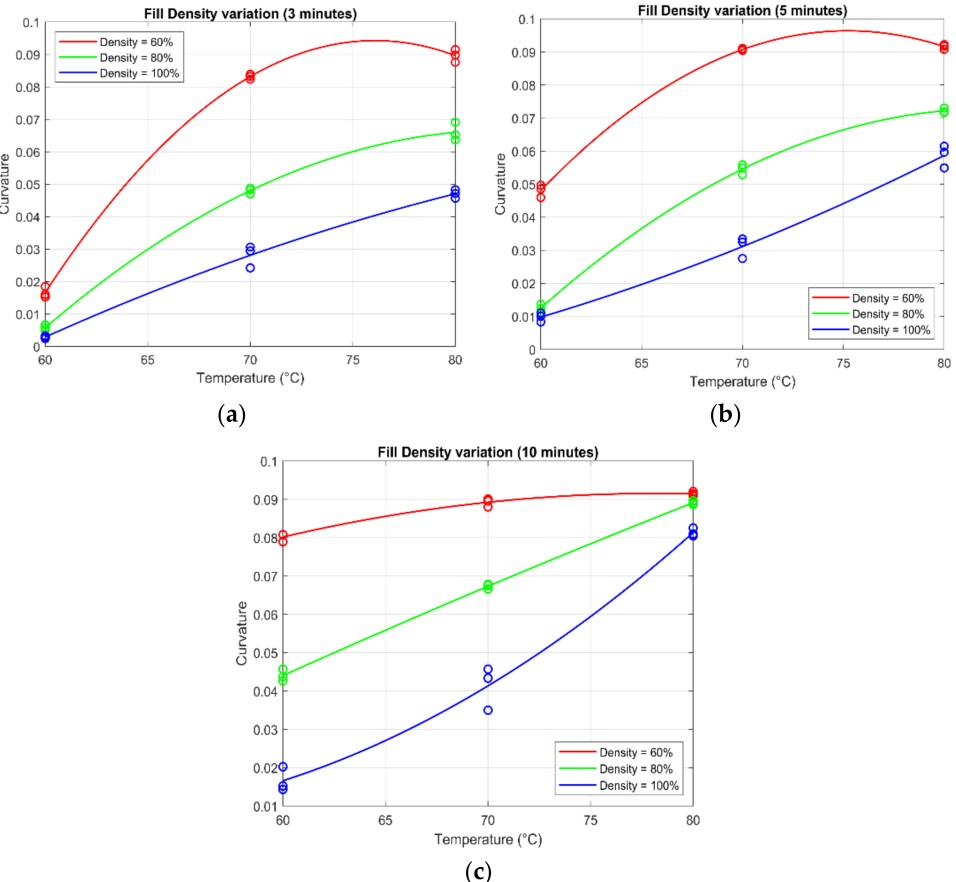

(**a**)

(**b**)

(**c**)

**Figure 10.** Curvature–temperature plot of fill-density variation (0.6 mm blue; 0.4 mm green; 0.2 mm red) for (**a**) 3, (**b**) 5, and (**c**) 10 min.

The plots in Figure 10 were obtained by curve fitting using a second-degree polynomial approximation with a goodness-of-fit rate of 95%. It is evident from the plots that the reduction in fill density allows for very pronounced curvature, even for a short exposure time of 3 min at a temperature of 60 °C (close to the glass transition temperature).

In addition, it is important to emphasize that the specimens exhibit pure flexural deformations without irregularities, unlike the specimens with small thicknesses. The equation that approximates any combination of curvature, as a function of temperature and time, as density changes has the same structure as Equation (2). Again, as in the previous case, due to the simplicity of the equation, for each exposure time and fill-density value, there are several fitting constants; consequently, in the following section, one of the

other two variables (fill density and time) is introduced into the fitting process and in the equation to highlight the variability in 3D plots.

### 3.4. Surface Fitting

The many numerical results make it possible to deepen the analysis of the data through a more complete fitting process considering more independent variables. Due to the complexity at the mathematical level, the step forward that has been taken is to include the geometric variable within the analysis, thus obtaining an equation that also considers thickness in one case, and fill density in another.

### 3.4.1. Thickness Variation (Surface Fitting)

By taking advantage of MATLAB's curve-fitting toolbox and appropriately grouping all the values of the independent variables and the obtained experimental results, it was possible to create curvature surfaces in space for each selected exposure time and approximate all the results via a second-degree polynomial function (for consistency with previous results). Below is an overall plot incorporating the three curvature surfaces (one per each exposure time) obtained for temperature and thickness variations (Figure 11). These surfaces are the graphical translation of the following numerical equation derived from the fitting process:

$$\kappa(T,\,h) = p_{00} + p_{10}T + p_{01}h + p_{20}T^2 + p_{11}Th + p_{02}h^2 \tag{3}$$

where $h$ indicates the thickness, $T$ the air temperature in degrees Celsius, $p_{ij}$ the fitting constants in 2D, and $\kappa$ the curvature. It is worth mentioning that, as each curvature surface refers to a different exposure time, it has different 2D fitting constants. In Table 4, the fitting constants of interest are grouped.

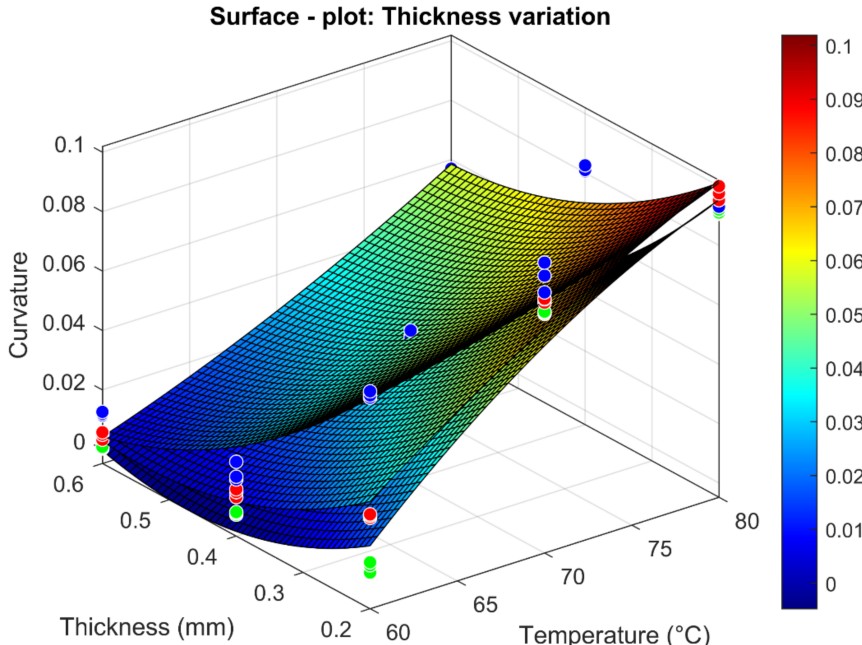

**Figure 11.** Curvature surface plot depending on the temperature (x-axis), the thickness (y-axis), and the exposure time (3 min (green), 5 min (red), and 10 min (blue). The colored histogram shows the numerical value of the curvature ranging from 0 to 0.1.

**Table 4.** Fitting constants per each exposure time (thickness variation).

| Time | Constants | | | | | |
|------|-----------|--------|--------|--------|--------|--------|
| | $p_{00}$ | $p_{10}$ | $p_{01}$ | $p_{20}$ | $p_{11}$ | $p_{02}$ |
| t = 3 min | −0.7 | 0.02 | 0.16 | $−9 \times 10^{-5}$ | −0.007 | 0.28 |
| t = 5 min | −0.4 | 0.01 | 0.03 | $−4 \times 10^{-5}$ | −0.005 | 0.25 |
| t = 10 min | 0.2 | −0.004 | −0.5 | $3.5 \times 10^{-5}$ | 0.002 | 0.25 |

All the considerations made in the previous paragraphs can be seen in the 3D graph. In addition, an analysis of the fitting constants obtained reveals a consistent change in the material's behavior when moving from 5 to 10 min of exposure time. The fitting constants present the same sign for the 3 and 5 min exposure times, and the opposite sign for the 10 min exposure time (except for the last term). The analogous analysis corresponding to the change in fill density is described below.

3.4.2. Fill-Density Variation (Surface Fitting)

As is the case with thickness variation, the overall graph incorporating the three curvature surfaces (one per exposure time) obtained as temperature and fill-density variation is shown in Figure 12. Again, the corresponding equation is a polynomial equation of the second degree, as shown below:

$$\kappa(T, \rho) = p_{00} + p_{10}T + p_{01}\rho + p_{20}T^2 + p_{11}T\rho + p_{02}\rho^2 \qquad (4)$$

where $\rho$ indicates the fill density. In such a case, it should be noted that the three curvature surfaces reflect Equation (4), but depending on a different exposure time, they have different fitting constants.

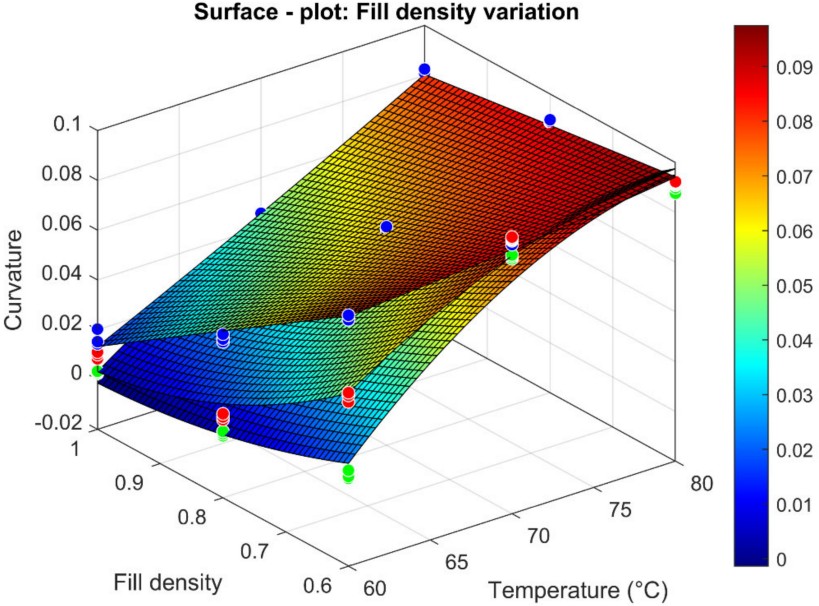

**Figure 12.** Curvature surface plot depending on the temperature (x-axis), the fill density (y-axis), and the exposure time (3 min (green), 5 min (red), and 10 min (blue)). The colored histogram shows the numerical value of the curvature ranging from 0 to 0.1.

In Table 5, the fitting constants of interest are grouped.

**Table 5.** Fitting constants per each exposure time (fill-density variation).

| Time | Constants | | | | | |
|---|---|---|---|---|---|---|
| | $p_{00}$ | $p_{10}$ | $p_{01}$ | $p_{20}$ | $p_{11}$ | $p_{02}$ |
| t = 3 min | −0.95 | 0.027 | −0.026 | $-1.5 \times 10^{-4}$ | −0.004 | 0.12 |
| t = 5 min | −0.35 | 0.016 | −0.5 | $-1 \times 10^{-4}$ | −0.001 | 0.21 |
| t = 10 min | 0.44 | −0.005 | −0.56 | $0.1 \times 10^{-4}$ | 0.007 | −0.0034 |

Compared to the case of thickness variation, the increase in curvature along the fill-density axis is very evident graphically, suggesting that this is the best parameter to vary to optimize the SME activation range. Furthermore, the analysis of the fitting constants obtained reveals a consistent change in the material's behavior when going from 5 to 10 min of exposure, just as in the previous case; in fact, the fitting constants present the same sign for exposure times of 3 and 5 min, and the opposite sign for the 10 min exposure time (except for the third term). This suggests that the fitting constants numerically depend on the exposure time. Notwithstanding, this mathematical pattern may suggest that the combination of the independent variables possesses a non-polynomial time-related term. The last case similar to the previous ones follows, where the geometric parameters are kept constant and time as an independent variable is introduced into the equation.

### 3.4.3. Passive Sensor (Time Variation)

This last case analysis was obtained by testing specimens with fibers arranged at 0° and 90°, with a total thickness of 0.4 mm and a fill density of 60%. The fitting process for this specimen was investigated in detail as, from the experimental results, it was the one that allowed the most significant optimization in terms of the SME activation range without incurring irregular deformations that could compromise the structure. By assuming constant geometric parameters, the curvature was fitted considering temperature and time as independent variables. Thus, only one surface resulted from the fitting process, as shown below in Figure 13. The curvature value for an exposure time of 3 min at 60 °C resulted in being approximately two times that of all other analyzed cases. Moreover, a decrease in the $\kappa$ value for an exposure time of 10 min at 80 °C was even more evident. The curvature reduction was probably due to a geometric constraint, since the maximum deflection cannot increase given the flap contact. However, it is also safe to assume that there exists a static equilibrium between all the entities involved, as no specimen out of all those analyzed showed a second generable curvature, regardless of time. As before, the reference equation is a second-degree polynomial, where, however, the independent variables are temperature and time (t):

$$\kappa(T,t) = p_{00} + p_{10}T + p_{01}t + p_{20}T^2 + p_{11}Tt + p_{02}t^2 \tag{5}$$

In addition, the following Table 6 shows the numerical values of the fitting constants involved.

**Table 6.** Fitting constants (time variation).

| Constants | $p_{00}$ | $p_{10}$ | $p_{01}$ | $p_{20}$ | $p_{11}$ | $p_{02}$ |
|---|---|---|---|---|---|---|
| Value | −1.17 | 0.030 | 0.042 | −0.0001 | −0.0004 | −0.0006 |

Compared with the previous cases, these multiplicative numerical constants have less significance. However, based on the found fitting constants, the quadratic contribution is smaller than the linear one. This may suggest that the nonlinear terms could be exponential or logarithmic. To fully understand the phenomenon mathematically, one would probably have to increase the amount of data available and combine all the independent variables (temperature, time, thickness, fill density, and gravity) into a single characteristic

numerical equation (not necessarily polynomial). This would allow for the separation of the contribution of SME and viscoelastic effects.

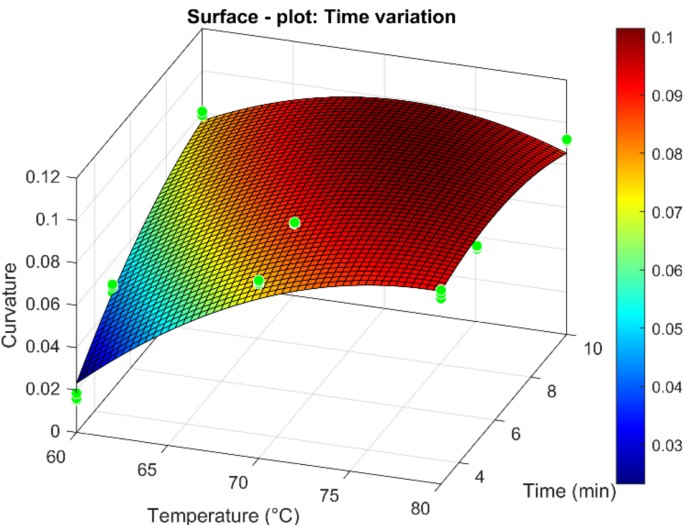

**Figure 13.** Curvature surface plot depending on the temperature (x-axis) and the exposure time variation (y-axis). The colored histogram shows the numerical value of the curvature ranging from 0 to 0.1.

## 4. Conclusions and Outlooks

Numerous results and considerations regarding material behavior and the shape-memory effect were obtained through the conducted experiments and analyses on 180 specimens performed in this research. However, by using curve- and surface-fitting tools for data analysis, the complexity of the problem was even more evident, given a large number of independent variables involved and the lack of mathematical knowledge related to SME. However, for the possible applications and feasible devices, such as passive sensors, the methodology of applied research proved to be the correct way to pursue the studies. It was ascertained that samples with a filling density of 60%, and a total thickness of 0.4 mm, responded readily to the activation of the shape-memory phenomenon, showing high flexural curvatures for short exposure times and for temperatures close to that of the glass transition (55 °C). Thereby, through this research, the potential of this new 4D-printing technology and the possible results that can be achieved with more in-depth future work is understood. To extend this research, it would be fascinating to interpolate all the experimental results obtained, looking for a link between the independent variables. Then, a characteristic constant of the SME could be derived, which links all the variables, and which could depend on the amount of PLA used to produce the component itself. All this could lead to an understanding about the amount of residual stress generated during the printing process. Continuing in this direction, through more excellent computer support, the development of a complete prediction of deformations (via FEA), and the implementation of the possibility of cyclic recovery of one's form, one could easily associate this technology with various fields of engineering application. The goal, given the high potential, could be the development of bio-based passive sensors that do not require external (electronic) components and reduce material waste and environmental impact due to the minimization of the number of components and the biodegradability of the PLA.

**Author Contributions:** Conceptualization, C.B.; Data curation, F.C.; Formal analysis, F.C.; Super-vision, C.G., F.P. and C.B.; Validation, M.M.; Writing—original draft, F.C.; Writing—review & edit-ing, M.M., C.G., F.B., F.P. and C.B. All authors have read and agreed to the published version of the manuscript.

**Funding:** This research received no external funding.

**Informed Consent Statement:** Not applicable.

**Conflicts of Interest:** The authors declare no conflict of interest.

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
