# Peer review of "Experimental Investigation of Thermal Passive-Reactive Sensors Using 4D-Printing and Shape-Memory Biopolymers"

_sustainability, doi:10.3390/su142214788_

Round 1

Reviewer 1 Report

Several issues must be addressed by the authors before publication consideration:

Please see the attached file containing the reviewer's revision.

Author Response

Dear Reviewer,
Attached are responses to all comments in a pdf document. Thank you very much 

Reviewer 2 Report

This manuscript described an interesting method for 4D printing technology related to the FDM using PLA. Via investigating the impactor factor like fiber direction, thickness, and fill density, the authors want to propose a method for the development of a sustainable and environmentally friendly passive sensor.

Overall, the goal is ambitious and the contents are well described. While the reviewer has some major questions during reviewing this manuscript: 1) The heating temperature and time DID trigger the SME, but how do they impact the SME, and how to utilize this quick/slow trigger to fabricate a passive sensor? I have an impression that the overall scope is big but the detailed plan has not been described clearly with strong connections. The author team needs to provide more information, to build a link about your goal and detailed plan in this manuscript. 2) About the term 4D printing, it seems that the time will ONLY impact the 3D printed works under certain conditions (like higher temperature). It is not like time directly works on the print result. Thus, I am not sure if it could be named as “4D printing” or pseudo-4D printing.

Other concerns/questions below arise from the reading of the paper, especially on the data/result analysis. Please address them in the revised work for the publication of this manuscript on Sustainability

 •           Line 104-105: Please draw an illustration to show the layout of those designed samples.

•           Line 122-123: How do you measure the maximum deflation and how to track the corresponding location? Do you have a camera to record the real-time change?

•           Figure 1: What’s the dimension information about your temporary shape

•           Figure 2: The black fiber pattern is in Layer 1 and the red fiber pattern is in Layer 2? Those figures are confusing and not like a two-layer structure. Recommend to use an isometric view/drawing to show two layers with different fiber filled-angles.

•           Figure 3 & Table 1: Mark the temperature information and sample information in figure 3A and 3B; or Combine your Figure 3 and Table 1 together, displaying the curvature value and image in a single cell.

•           Figure 5: Similar comments as Figure 3’s. Mark the temperature information and sample information in figure 5A-C and make the caption with clear and full information.

•           Line 193: Are those three types of samples bi-layered structures with total thicknesses of 0.2mm, 0.4mm, and 0.6mm? Does it mean that the single layer in each sample is 0.1mm, 0.2mm, and 0.3mm?

•           Line 205: What do you mean by saying “First row, bottom part”? If it’s the first row, where is the bottom part come from? It’s better to mark Fig 5A1-5A9, to help the readers and reviewers to locate the sample you discussed in your context.

•           Line 206: “the specimens with a total thickness of 0.2mm (first row, bottom part) for high time-temperature values (e.g., 10 min at 80 °C) show irregularities in the final shape” From your image, it seems the specimen in Figure C, top row, 3rd column should be the one you talk about. And it has the largest curvature. What are the irregularities? Does it matter? 

•           Line 235: What’s your justification for choosing the specimens with a thickness=0.2mm to compare in this section? From your section 3.2, it seems that the specimens with a total thickness of 0.2mm were not optimal. Will this selection impact your following result and analysis?

•           Figure 7: What’s the grey part? Are you printing three different types of specimens with varied fill rates on the same “Grey” base layers? That figure is not clear to illustrate your three types of samples. Please make a proper modification.

•           Line 250: What do you mean by saying “greater curvatures are achieved than the thickness variation”? From your previous tests, increasing the thicknesses of samples will retard gaining the greater curvatures but you haven’t tested in the order direction, reducing the thickness. Thus here, I don’t understand why you have such a claim.

•           Line 281: “temperature and time is identical to eq. 2.” In your Equation 2, the varied factor is T (Temperature) and the constant factors are related to time and thickness. So here “Temperature” should not be the same to eq 2.

•           Equation 3: How to control your fill density as the constant? In your equation 3 and Fitting constants, I cannot see the impact of the fill density.

•           Equation 4: Same as the comments to Equation 3, you need to provide the impact from the thickness. And also if you have made the thickness (here) or fill density (in Eq 3) constant by default. 

Author Response

(The authors gave the same response as above.)

Reviewer 3 Report

This paper proposed a strategy for fabricating 4D-printed sensors using shape memory polymer PLA and FDM printing method. The sensors can perform bending deformations by the design of geometric parameters. Sufficient experimental data and theoretical analysis are provided to prove the influence of parameters such as structure, temperature and time on deformation effect. I recommend acceptance of this paper for publication after minor revision, with the following comments properly addressed,

1.     The discussion on related work is not concise and condensed enough, and the innovation and significance of this work is not clear enough.

2.     Please check the format. There should be spaces between numbers and units.

3.     Figure 2 and the corresponding explanations are confusing. The thickness of the sample single layer is 0.2mm, what about the total thickness? How many layers is the sample? Do different filling angles correspond to different layers of film or different positions within a single layer of film?

4.     Pay attention to the specification of the units. In Table 1 and Table 2, the abbreviation for minutes should be min instead of m.

5.     Figures need to be clearer and more aesthetically pleasing. Proper adjustments to the layout and proper labeling on the drawing are required.

Author Response

(The authors gave the same response as above.)

Round 2

Reviewer 1 Report

The authors have properly edited the manuscript and followed the reviewer's suggestions. Nevertheless, few minor issues need to be addressed before publication.

1- the abstract should contain a brief description of the main results observed. It could show some values and/or mathematical correlations (e.g. x fold greater, etc.)

2- extensive phrases (with 4-5 lines) should be avoided to enhance the reader's comprehension. More concessive sentences should be given in the following sections (e.g. lines 138-142, 503-507, 560-564)

3- the authors stated that the main motivation of the manuscript is given in the conclusion section (see line 138). The paragraph is not clear enough. The authors are encouraged to rephrase this sentence and clearly demonstrate the objective in the introduction section.

4- in the previous version of the manuscript the authors stated:

"R1.4 -In lines 39-40, what are the common materials/polymers utilized in the manufacturing of referred systems?

A1.4 -Checked"

The information (regarding the answer to R1.4) is missing in the current version of the manuscript. Please add it.

Author Response

Dear reviewer, attached are the responses to your comments/corrections; thank you very much
